# Robust Topological Edge States in $C_6$ Photonic Crystals

**Daniel Borges-Silva** [1,2] [iD], **Carlos Humberto Oliveira Costa** [3] [iD] and **Claudionor Gomes Bezerra** [2,*] [iD]

1 Instituto Federal do Ceará, Cedro 63400-000, Ceará, Brazil; daniel.silva@ifce.edu.br
2 Departamento de Física, Universidade Federal do Rio Grande do Norte, Natal 59078-900, Rio Grande do Norte, Brazil
3 LAREB, Universidade Federal do Ceará, Russas 62900-000, Ceará, Brazil; carloshumberto@ufc.br
* Correspondence: cbezerra@fisica.ufrn.br

**Abstract:** The study of photonic crystals has emerged as an attractive area of research in nanoscience in the last years. In this work, we study the properties of a two-dimensional photonic crystal composed of dielectric rods. The unit cell of the system is composed of six rods organized on the sites of a $C_6$ triangular lattice. We induce a topological phase by introducing an angular perturbation $\phi$ in the pristine system. The topology of the system is then determined by using the so-called **k.p** perturbed model. Our results show that the system presents a topological and a trivial phase, depending on the sign of the angular perturbation $\phi$. The topological character of the system is probed by evaluating the electromagnetic energy density and analyzing its distribution in the real space, in particular on the maximal Wyckoff points. We also find two edge modes at the interface between the trivial and topological photonic crystals, which present a pseudospin topological behavior. By applying the bulk-edge correspondence, we study the pseudospin edge modes and conclude that they are robust against defects, disorder and reflection. Moreover, the localization of the edge modes leads to the confinement of light and the interface behaves as a waveguide for the propagation of electromagnetic waves. Finally, we show that the two edge modes present energy flux propagating in opposite directions, which is the photonic analogue of the quantum spin Hall effect.

**Keywords:** topological photonic crystal; edge states; $C_6$ symmetry group; electromagnetic density energy





## 1. Introduction

Photonic crystals (PCs) are systems whose electromagnetic features periodically modulate in the real space [1]. In particular, two-dimensional (2D) photonic crystals can present topological behavior by introducing a perturbation in the Hamiltonian of the system in order to obtain a non-zero Chern number [2,3]. Two-dimensional topological photonic crystals are currently in the scientific limelight not only for possessing tremendous technological potential but also for having opened several avenues of basic science exploration [4–11]. It is known that when the photonic crystal is in a nontrivial topological phase, interesting phenomena and properties can emerge. For example, we can highlight photonic analogies of the quantum Hall effect [12,13], higher-order topological photonic crystals [14–16], the quantum valley Hall effect [17–20], Huygens–Fresnel electromagnetic transportation [21], fano resonance [22,23], antichiral states [24], and anisotropic topological phases [25]. Those unique physical properties make photonic crystals excellent candidates for potential technological applications, such as topological lasers [26,27], topological waveguides [28–30], filters, and resonators [31–33].

Furthermore, all these topological properties can be engineered in PCs. Changes in the structural configuration, such as extrinsic defects and imperfections, can modify the topological properties of the crystal, leading to topological phase transitions, and create localized states. As a matter of fact, controlled perturbations can significantly impact the formation of topology in PCs. For example, one can tune physical parameters such as

lattice spacing, refraction index, spacial orientation of the "artificial atoms", etc., inducing reconfiguration of topological edge states which may be potential for reconfigurable topological waveguides [34]. In particular, it is known from the literature that photonic crystals in a triangular lattice, with $C_6$ point symmetry group, present a doubly degenerate Dirac cone at the $\Gamma$ point of the Brillouin zone. Moreover, the degenerate bands present $p-$ and $d-$ wave orbitals in the band structure of the transverse magnetic (TM) polarization [35,36]. However, if a perturbation is introduced in the crystal, the double degenerescence is broken and a complete topological bandgap is opened in the band structure [37]. In addition, the bulk-edge correspondence guarantees that edge states will emerge inside the bandgap and they will be protected by the topology [38,39]. Then, the topological protection ensures that the edge states are localized and that they are robust against defects, disorder, and reflection [40,41].

In this work, we investigate the propagation of electromagnetic waves, band structure, and topological features of a two-dimensional topological photonic crystal composed of six dielectric cusped-oval-shaped (COS) rods. We induce a bandgap in the system by introducing a perturbation in the rods' orientation angle, lifting the double degeneracy at the $\Gamma$ point of the Brillouin zone. In the following, we will show that two edge modes emerge in the induced bandgap. This paper is organized as follows. In Section 2, we introduce our system and explore its features. In Section 3, we introduce the perturbation and study the topological behavior associated with positive and negative perturbations. In Section 4, we study the emergence of edge states around the interface between the topological and trivial photonic crystals. The robustness of the edge states is addressed in Section 5. Finally, in Section 6 we give concluding remarks about this work.

## 2. The Photonic System

In order to obtain the band structure of the 2D PC considered here, we must study the light dynamics in the system. It is well known from the literature that light dynamics in photonic crystals comes from the Maxwell equations [1], with the time-dependent fields given by $\vec{E}(\vec{r}, t) = \vec{E}(\vec{r})e^{i\omega t}$ and $\vec{H}(\vec{r}, t) = \vec{H}(\vec{r})e^{i\omega t}$. For non-magnetic materials $\mu(\vec{r}) = 1$ and with no free charge and current density, the equation for the electric field, the Master Equation, may be written as

$$\frac{1}{\varepsilon(\vec{r})} \nabla \times [\nabla \times \vec{E}(\vec{r})] = \frac{\omega^2}{c^2} \vec{E}(\vec{r}). \tag{1}$$

Once the electric field's dynamics is found, we can obtain the magnetic field's dynamics by using Faraday's relation $\vec{H}(\vec{r}) = -[i/\mu_0\omega]\nabla \times \vec{E}(\vec{r})$.

Consider now a triangular lattice with six dielectric rods ($\varepsilon = 13$) per site surrounded by air. The unit cell is composed of dielectric COS rods, with $a = 1$ μm, as we can see in Figure 1. The COS rods are built taking into account two cylindrical rods of radius $r = 0.17a$, which are shifted by a distance $d = \pm a/15$. Then, we consider the intersection area between the shifted cylindrical rods. Finally, the COS rods are placed in their locations which are distant $R = a/3$ from the unit cell's center (see Figure 1), in order to obtain a 2D system with $C_6$ symmetry point group.

Each individual COS rod in the unit cell can be rotated around its respective center by the orientation angle $\Phi_i$, as illustrated in Figure 1. The rods present an anisotropic angular orientation which can vary spatially. Therefore, we may expect that an angular perturbation will lead the system to undergo topological phase transitions from trivial to nontrivial domain [42]. A perturbation $\phi$ is introduced in the orientation angle $\Phi_i$, so that we can write the orientation angle of the $i$-th COS rod as

$$\Phi_i = (2i - 1)\frac{\pi}{6} + \phi_0 + \phi. \tag{2}$$

Here, $i = 1, 2, ..., 6$ is the rod index, $\phi_0$ is the initial unperturbed angle and $\phi$ is the angular perturbation introduced in our system. We remark that, because of the symmetry,

the perturbation $\phi$ has a period $\pi$, which means that the $\phi$ and $\phi + \pi$ induce the same topological behavior in the system. Here, we assume $\phi_0 = \pi/4$ (see Figure 2a).

It is well known from the literature that triangular lattices with six "artificial atoms" have two 2D irreducible representations in the $C_6$ point symmetry group, which are associated with the symmetry of the triangular lattice [43]. As a consequence, a doubly degenerate Dirac cone appears at the Brillouin zone center because of the Kramers' degeneracy theorem as discussed below [44]. The degenerate bands are pseudospin states which are related to $p_x$ ($p_y$) and $d_{xy}$ ($d_{x^2-y^2}$) orbitals, corresponding to odd and even parity in the real space, respectively [35]. The pseudospin states can be written as [42]

$$p_\pm = \frac{1}{\sqrt{2}}(p_x \pm ip_y), \qquad d_\pm = \frac{1}{\sqrt{2}}(d_{x^2-y^2} \pm id_{xy}). \tag{3}$$

The two irreducible representations $D'(C_6)$ and $D''(C_6)$, with basis $(p_x, p_y)$ and $(d_{xy}, d_{x^2-y^2})$, respectively, allow us to write the pseudo-time-reversal operator $\mathcal{T} = \mathcal{U}\mathcal{K}$ in an invariant form. $\mathcal{K}$ is the complex conjugate operator and $\mathcal{U}$ can be defined by [35]

$$\mathcal{U} = \frac{1}{\sqrt{3}}[D'(C_6) + D'(C_6^2)] = \frac{1}{\sqrt{3}}[D''(C_6) - D''(C_6^2)]. \tag{4}$$

Here, $D'(C_6)$ is equivalent to a rotation of $\pi/3$, while $D''(C_6)$ is equivalent to a rotation of $2\pi/3$, i.e.,

$$D'(C_6)\begin{pmatrix} p_x \\ p_y \end{pmatrix} = \begin{pmatrix} 1/2 & -\sqrt{3}/2 \\ \sqrt{3}/2 & 1/2 \end{pmatrix}\begin{pmatrix} p_x \\ p_y \end{pmatrix} \tag{5}$$

and

$$D''(C_6)\begin{pmatrix} d_{x^2-y^2} \\ d_{xy} \end{pmatrix} = \begin{pmatrix} -1/2 & -\sqrt{3}/2 \\ \sqrt{3}/2 & -1/2 \end{pmatrix}\begin{pmatrix} d_{x^2-y^2} \\ d_{xy} \end{pmatrix}. \tag{6}$$

We should comment that once the system presents time-reversal-symmetry and behaves like a half-integer spin system [45,46], according to Kramers' degeneracy theorem, degenerescence spontaneously emerges in a similar way as it occurs in electronic systems [47,48].

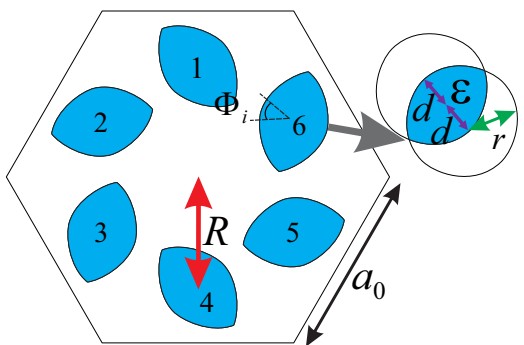

**Figure 1.** Schematic illustration of the unit cell of the unperturbed PC composed of six COS Si rods surrounded by air. Here, $a_0 = a/\sqrt{3}$, $R = a/3$ is the distance from the center of the unit cell to the center of the rods, $r = 0.17a$ is the radius of the original cylindrical rods, and $d = a/15$ is the shift of the original rods.

Considering the unperturbed photonic crystal ($\phi = 0$), we calculate the band structure for the TM modes ($E_z$, $H_x$, $H_y \neq 0$) using the COMSOL Multiphysics software [49] which is based on the finite element method (FEM). The band structure is shown in Figure 2a. We can observe a doubly degenerate Dirac cone, at the $\Gamma$ point, between the second and fifth bands, which is a consequence of the $C_6$ symmetry group of the system.

In Figure 2b, we plot the electric field along the *z* direction ($E_z$), at the Dirac point, with $\omega a/(2\pi c) = 0.437$. We found four states that are related to dipole and quadrupole modes. More specifically, $p_x$ and $p_y$ orbitals are dipole modes, while $d_{xy}$ and $d_{x^2-y^2}$ orbitals are quadrupole modes. In the next section, we introduce a nonzero perturbation in order to lift the double degenerescence and induce a complete photonic bandgap in the band structure.

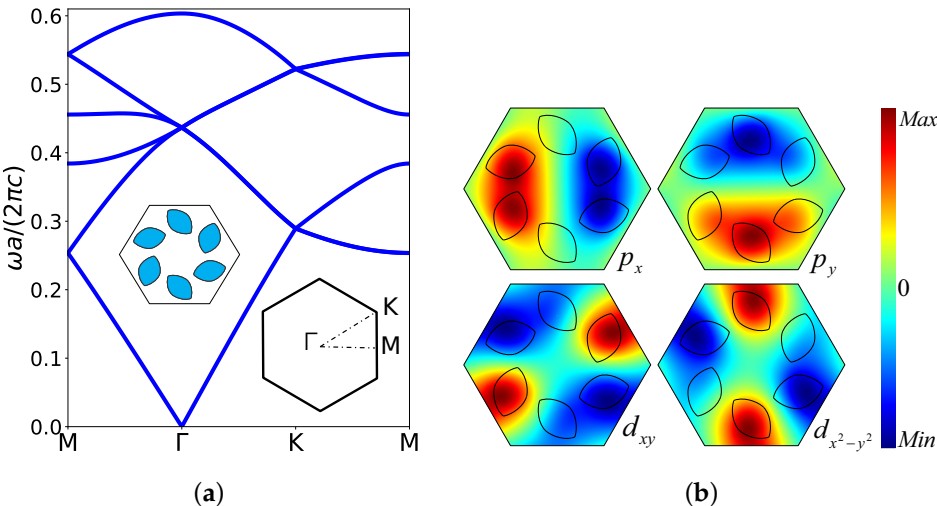

(**a**)             (**b**)

**Figure 2.** Results for the unperturbed photonic crystal. (**a**) Band structure for the TM modes with $\varepsilon = 13$, $a = 1$ μm, $r = 0.17a$, $d = a/15$, $\phi_0 = \pi/4$, and $\phi = 0$. A doubly degenerate Dirac point is located at the $\Gamma$ point with $\omega a/(2\pi c) = 0.437$. (**b**) Profile of $E_z$ at the Dirac point. We can see that the orbitals are dipole modes ($p_x$ and $p_y$), and quadrupole modes ($d_{xy}$ and $d_{x^2-y^2}$).

## 3. Topological Phase Transition

In Section 2, we have found a doubly degenerate Dirac cone at the $\Gamma$ point, as well as orbitals *p*-like and *d*-like which are associated to the degenerate bands. Let us now study the consequences of considering a nonzero perturbation $\phi$ in the rods' orientation angle. In order to illustrate the effects of the angular perturbation, we evaluate $\omega a/(2\pi c)$ *vs* $\phi$ for the second, third, fourth, and fifth bands, as we can see in Figure 3. We can observe that as $|\phi|$ increases from 0, the bandgap width monotonically increases, reaching its maximum at $\phi = -\pi/4$ and $\phi = \pi/4$. On the contrary, as $|\phi|$ increases from $\phi = -\pi/4$ and $\phi = \pi/4$, the bandgap width monotonically decreases until the doubly degenerate Dirac cone is recovered at $\phi = -\pi/2$ and $\phi = \pi/2$. This is a consequence, as mentioned before, of the angular perturbation $\phi$ having a period $\pi$.

We next illustrate the opening of the bandgap in the band structure for two values of $\phi$, one positive and other negative. Figure 4 shows the band structure corresponding to $\phi = -\pi/4$ and $\phi = \pi/4$. The perturbation opens a gap between $\omega a/(2\pi c) = 0.432$ and $\omega a/(2\pi c) = 0.4408$, for the positive case, and between $\omega, a/(2\pi, c) = 0.431$, and $\omega a/(2\pi c) = 0.4415$, for the negative case, corresponding to a gap–midgap ratio [1] of $\Delta\omega/\omega_m = 0.0202$ and $\Delta\omega/\omega_m = 0.0241$, respectively. One can observe that a complete bandgap is opened for both cases. Note that once the bandgap is opened, edge states can appear inside the gap, as we will see later in this paper. It is also interesting to observe in Figure 3 that the bandgap width is not symmetric in relation to $\phi$. Photonic systems can be mapped in tight-binding models [34,50]. Bearing this information in mind, observe in Figure 4 the unit cell configuration for $\phi = -\pi/4$ and $\phi = \pi/4$. The configuration for $\phi = -\pi/4$ corresponds to a tight-binding model with $t_0 > t_1$, where $t_0$, $t_1 > 0$, represent the nearest-neighbor (NN) hopping integrals inside and between the hexagonal unit cells. On the other hand, the configuration for $\phi = \pi/4$ corresponds to a tight-binding model with $t_0 > t_1 > 0$ and $t_2 \neq 0$. Here, $t_2 \neq 0$ represents the next-nearest-neighbor (NNN) hopping integral inside the diagonal unit cell. Once both configurations correspond to different tight-binding Hamiltonians, the gap width is different [34,50].

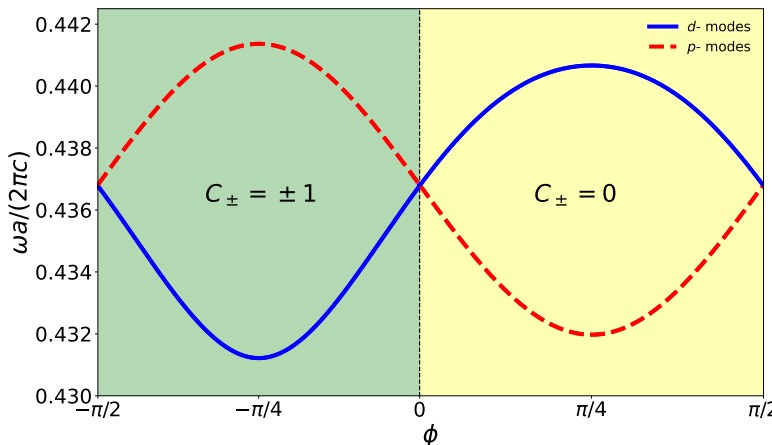

**Figure 3.** Effect of the angular perturbation $\phi$ on the TM band structure at the $\Gamma$ point. It is possible to observe that the double degeneracy is lifted when we introduce a nonzero perturbation $\phi$ (see the main text).

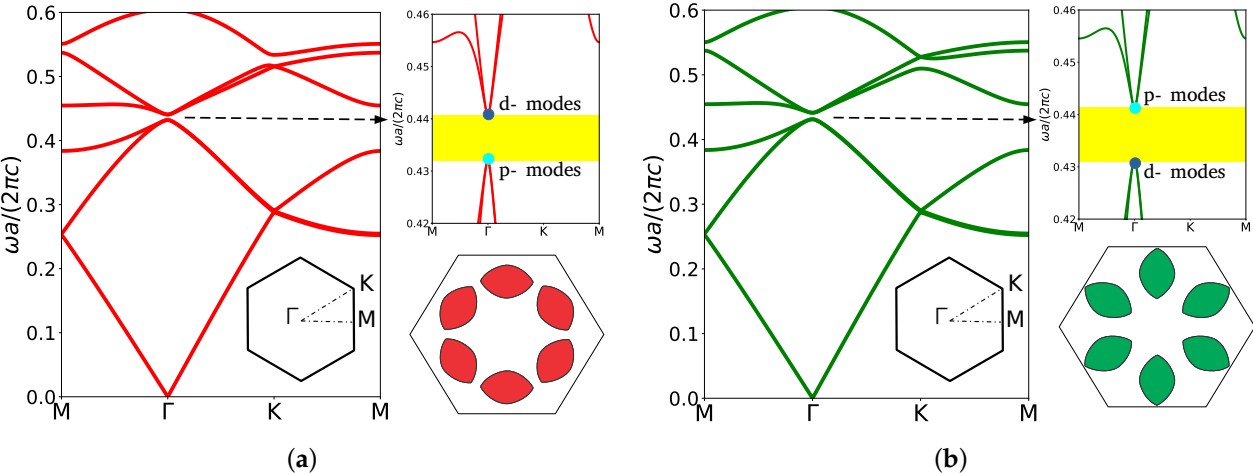

**Figure 4.** Results for the perturbed photonic crystal. Band structure of the TM modes with $\varepsilon = 13$, $a = 1$ μm, $R = a/3$, $r = 0.17a$, and $d = a/15$, for (**a**) the negative case illustrated by the red rods ($\phi = -\pi/4$) and (**b**) the positive case illustrated by the green rods ($\phi = \pi/4$), respectively. The bandgap is highlighted by the yellow area.

In order to study the topological behavior close to the $\Gamma$ point, we can write an effective Hamiltonian by using the **k.p** perturbed model from which we can obtain the Chern number [51–53]

$$C_{\pm} = \pm\frac{1}{2}[\text{sign}(B) + \text{sign}(M)]. \tag{7}$$

Here, $B$ is the diagonal term of the effective Hamiltonian close to the $\Gamma$ point which is essentially negative. In addition, $M = (\omega_d - \omega_p)/2$, where $\omega_d$ and $\omega_p$ are the eigenmodes of orbit $d$ and orbit $p$, respectively [54]. The eigenmode $\omega_p$ is related to the double degenerate dipole states of $p_{\pm}$, while $\omega_d$ is related to the double degenerate quadrupole states of $d_{\pm}$ [51]. If $\omega_p < \omega_d$ then $M > 0$, hence $C_{\pm} = 0$ and the photonic crystal is topologically trivial. However, if $\omega_p > \omega_d$ then $M < 0$, hence $C_{\pm} = \pm 1$ and the photonic crystal is in a topological phase. Therefore, the inversion of the bands between the degenerate modes at the $\Gamma$ point leads to the topological phase transition [55].

For this work, we consider the inversion of the bands that occurs between $\phi = -\pi/4$ and $\phi = \pi/4$ for the degenerate modes, as shown in Figure 5. We can observe that for the positive case the frequency of the dipole modes is lower than the frequency of the quadrupole modes, while for the negative case the frequency of the dipole modes is higher

than the frequency of the quadrupole modes. Therefore, for $\phi = \pi/4$ we obtain $C_\pm = 0$, which corresponds to a trivial photonic crystal, and for the case $\phi = -\pi/4$ we obtain $C_\pm = \pm 1$, which corresponds to a topological photonic crystal.

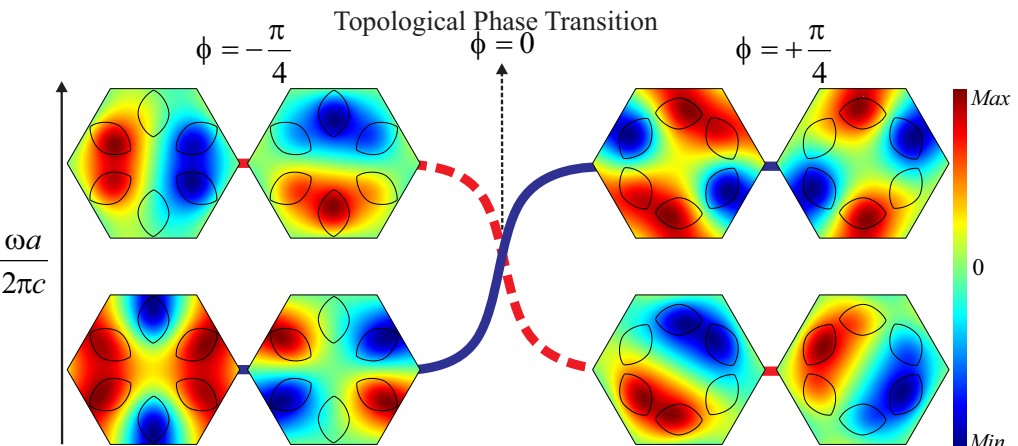

**Figure 5.** Topological phase transition diagram. Profile of the electric field $E_z$ of the degenerate bands. There is an inversion of the bands between $\phi = -\pi/4$ and $\phi = \pi/4$. The **left side** represents the topological case, and the **right side** represents the trivial case. The topological phase transition occurs when $\phi = 0$. The red and blue lines are a guide to the eyes.

We can obtain important information about the topological behavior of the photonic bands from the electromagnetic (EM) energy density distribution in the real space. The general idea is that the EM energy density has peaks that are shifted towards the maximal localized Wyckoff points (WP) at which the Wannier functions (WF) of the system are centered. This is a consequence of the relationship between the Wilson-loop (WL) operator and the maximally-localized WF. The spectrum of the WL operator is a useful method for characterizing the topological phases of physical systems. First, we write the WL operator as a path-ordered exponential of the Berry phase which is defined by [56]

$$\mathcal{W}_{mn}(l) = \mathcal{P}e^{-i\int_l \mathbf{A}_{mn}(\mathbf{k}).d\mathbf{l}}. \tag{8}$$

Here, $\mathcal{P}$ is the path ordering operator, and $\mathbf{A}_{mn}$ is the Berry connection for $m = n$. Second, it is known from the literature that there is a connection between the WL operator and the WF. The WF, which is defined as a Fourier transformation of the Bloch states, can be written as [57]

$$w_{i\mathbf{R}}(\mathbf{r}) \propto \int_{BZ} e^{-i\mathbf{k}.\mathbf{R}} \sum_j U_{ij}^{\mathbf{k}} \psi_{j\mathbf{k}}(\mathbf{r}). \tag{9}$$

Here, $U_{ij}^{\mathbf{k}}$ denotes the mixing matrix, which represents the mixing of the Bloch modes in the reciprocal space.

When we take into account the maximally localized WF, the mixing matrix takes values to minimize the delocalization of the wave function according to the eigenvalue of the WL operator [58]. The sum of the phases of the operator's eigenvalues provides a straight line in the Brillouin zone that corresponds to the expectation value of the projected position operator calculated over the maximally localized WF [59]. For trivial systems, the WL spectrum does not present winds and the Chern number is zero. Moreover, the maximally localized Wannier functions are *exponentially localized* in specific points of the real space. On the other hand, for topological systems the WL eigenvalues present winds and the maximally localized WFs are not exponentially localized, but *polynomially localized* in the real space between consecutive unit cells [57]. Therefore, we can identify different topological phases by looking for the positions of the maxima of the WF in the real space.

As mentioned before, the peaks of the EM energy density are shifted towards the maximal localized WP at which the WF are centered. Therefore, the EM density energy

can be a useful tool to probe the topological features of the system. We can obtain the EM energy density from the local density of states for a set of connected bands $\Lambda$ and considering the TM polarization as [57]

$$n_\Lambda(\mathbf{r}) = \frac{6}{\pi S} \sum_{n \in \Lambda} \int_{BZ} d\mathbf{k} \varepsilon(\mathbf{r}) \mid \mathbf{E}_{\mathbf{k}n}(\mathbf{r}) \mid^2 . \tag{10}$$

Here, $S$ is the area of the unit cell in the real space and $\mathbf{E}_{\mathbf{k}n}(\mathbf{r})$ is the electric field of the $n$-th band. It is possible to find the total EM energy density by summing all sets of bands, i.e., $n(\mathbf{r}) = \sum_\Lambda n_\Lambda(\mathbf{r})$. Moreover, the total energy density is written in terms of the electromagnetic field WF, i.e., $\mathbf{E}_{\mathbf{k}n}^w(\mathbf{r})$ [60]:

$$n(\mathbf{r}) = \frac{6}{\pi S} \sum_\Lambda \sum_{n \in \Lambda} \sum_\mathbf{R} \varepsilon(\mathbf{r}) \mid \mathbf{E}_{\mathbf{k}n}^w(\mathbf{r}) \mid^2 . \tag{11}$$

Since the WFs are linked to the WL, Equation (11) allows us to indirectly identify the topological behavior by calculating the EM density and study its maximal localization in the real space without directly evaluating the WL [57].

For the photonic crystal considered in this work, we evaluate the EM energy density distribution for the positive and negative perturbation cases ($\phi = \pi/4$ and $\phi = -\pi/4$, respectively), as we can see in Figure 6. From Equation (10), we can notice that the energy density depends on the permeability parameter $\varepsilon(\mathbf{r})$, which assumes the values $\varepsilon(\mathbf{r}) = 1$ and $\varepsilon(\mathbf{r}) = \varepsilon$ for the background and rods, respectively. The difference between these two values of permeability makes the intensity of the EM density in the rods much higher than in the background. Thus, evaluating this quantity in the entire unit cell does not result in trustworthy data for the EM density energy localization in the background. In order to circumvent this problem, we separately evaluate the energy density for the rods and for the background. For the latter case, we consider $\varepsilon \to 1$ for the rods.

Since the topological behavior of a bandgap can be defined as the sum of the topological behavior of the bands below that bandgap [8], we can focus on the set of bands below the bandgap at the $\Gamma$ point (see Figure 4). Figure 6 shows the EM energy density distribution in the unit cell and the maximal WP. In particular, results in the literature show that topological phases tend to present the associated EM energy density around the $3c$ WP in the edge of the unit cell [57]. From Figure 6a, we conclude that the first band does not contribute to the topological features of the bandgap since it has a homogeneous energy distribution. Therefore, we can focus on the other bands below the bandgap, i.e., the second and third bands (see Figure 4a). As expected, we see that the maximal EM energy density in the rods is located in the regions around the $3c$ WP at the edge of the unit cell. The same behavior is observed for the background, but with the maximum EM energy density located right on the $3c$ WP. Therefore, we can infer that the negative perturbation leads our system to a nontrivial topological phase. On the other hand, in Figure 6b all the bands below the bandgap, i.e., the first, second, and third bands (see Figure 4b) contribute to the topological features. We can observe that the maximum EM energy density is localized inside the rods but far from the maximal WP. Focusing on the background, we notice that the maximum is located between the rods, in the middle of the distance between the $1a$ WP and the edge of the unit cell. Thus, for the positive case, the perturbation leads the system to a trivial topological phase.

Let us quantify the localization illustrated in Figure 6. In order to do so, we set two lines: (i) one along the direction $(\mathbf{a}_1 + \mathbf{a}_2)$ and (ii) another along the direction $(\mathbf{a}_1 - \mathbf{a}_2)$. Next, we evaluate the EM energy density along those lines as shown in Figure 7. Figure 7a,c show the energy density $n_\Lambda / \max[n_\Lambda]$ along the $(\mathbf{a}_1 - \mathbf{a}_2)$ and $(\mathbf{a}_1 + \mathbf{a}_2)$ directions, respectively, for $\phi = -\pi/4$. On the other hand, Figure 7b,d show the energy density $n_\Lambda / \max[n_\Lambda]$ along the $(\mathbf{a}_1 - \mathbf{a}_2)$ and $(\mathbf{a}_1 + \mathbf{a}_2)$ directions, respectively, for $\phi = \pi/4$.

Analyzing Figure 7, we observe that bands 2 and 3, for the negative case, have the maximal EM energy density at the $3c$ WPs (red-solid line in Figure 7a,c), corresponding

to *polynomially localized* WF. On the other hand, for the positive case, bands 1–3 have the maximal EM energy density located in the region either between 1*a* and 2*b* WPs or between 1*a* and 3*c* WPs (red-solid line in Figure 7b,d), corresponding to *exponentially localized* WF. We can infer that the results about localization of the EM energy density reinforce our conclusion about the topological behavior for the negative perturbation case and the trivial behavior for the positive perturbation case. It is important to highlight that the results illustrated in Figure 7 are in complete agreement with the previous results obtained from the **k.p** perturbed model, which we used to evaluate the Chern number.

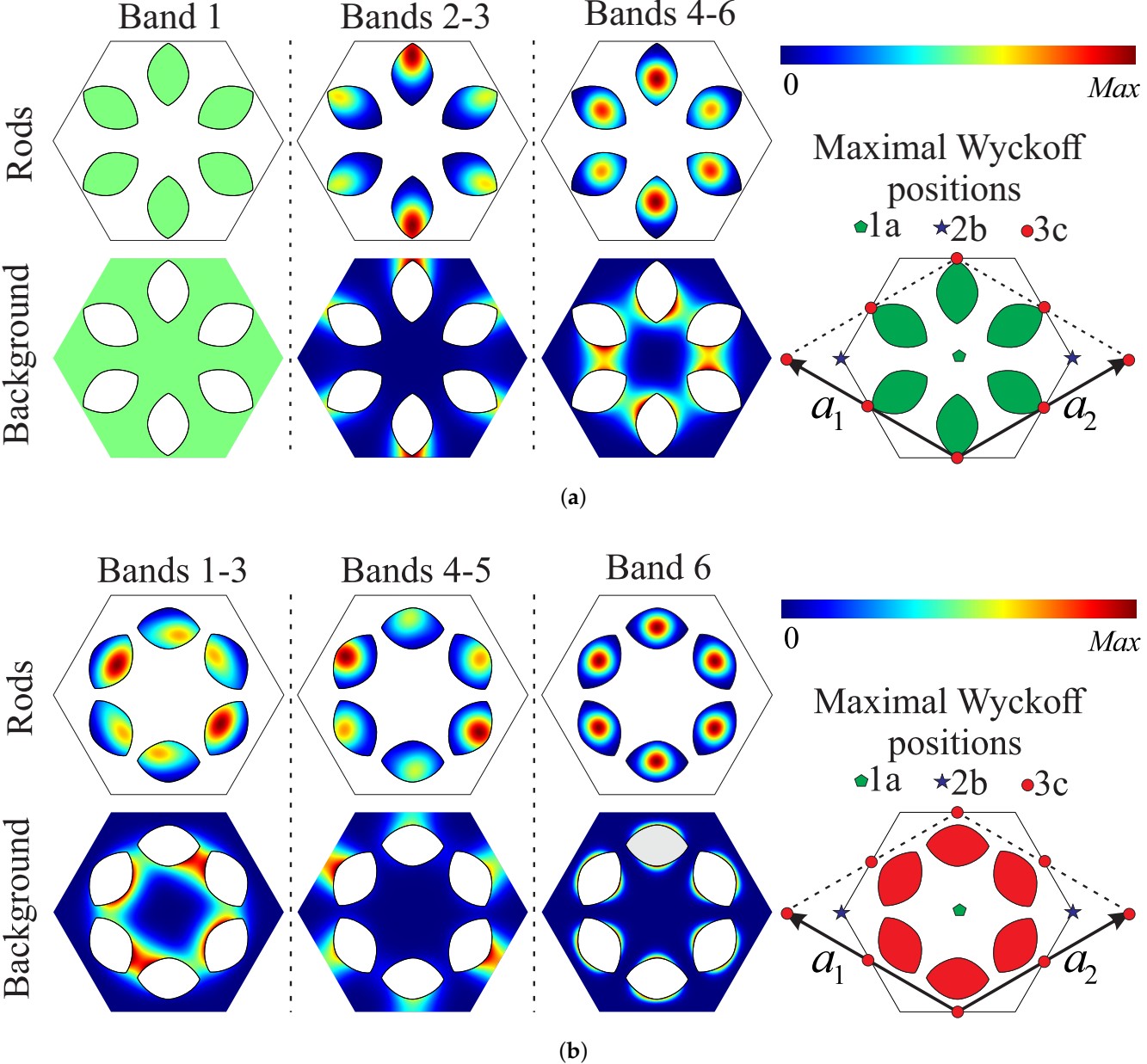

**Figure 6.** Energy density distribution for TM modes with $\varepsilon = 13$, $a = 1$ μm, $R = a/3$, $r = 0.17a$, and $d = a/15$ for the (**a**) negative case ($\phi = -\pi/4$), and (**b**) positive case ($\phi = \pi/4$). The positions of the maximal WP 1*a*, 2*b* and 3*c* are also illustrated.

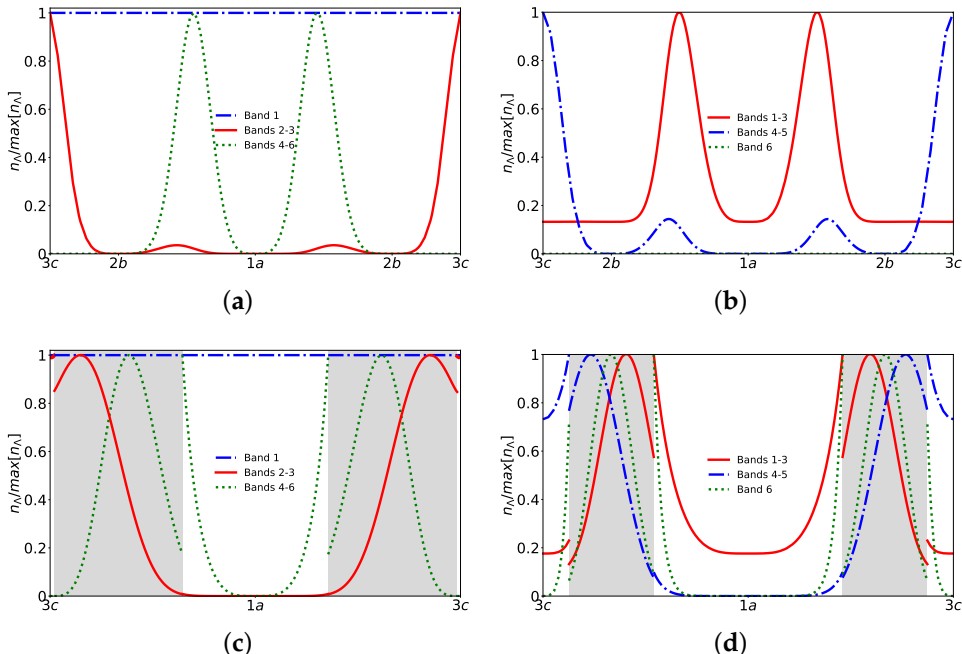

**Figure 7.** $n_\Lambda / \max[n_\Lambda]$ for TM modes with $\varepsilon = 13$, $a = 1$ μm, $R = a/3$, $r = 0.17a$, and $d = a/15$ for (**a**) $\phi = -\pi/4$ along the direction $(\mathbf{a}_1 - \mathbf{a}_2)$, (**b**) $\phi = \pi/4$ along the direction $(\mathbf{a}_1 - \mathbf{a}_2)$, (**c**) $\phi = -\pi/4$ along the direction $(\mathbf{a}_1 + \mathbf{a}_2)$, (**d**) $\phi = \pi/4$ along the direction $(\mathbf{a}_1 + \mathbf{a}_2)$. The gray areas denote the rods.

## 4. Edge States

We have shown that the perturbation $\phi$ opens a complete bandgap at $\Gamma$ point and we can identify two different phases, the topological phase for $\phi < 0$ and a trivial phase for $\phi > 0$. On the other hand, it is known from the literature that the bulk-edge correspondence guarantees that if we build a slab composed of two photonic crystals, with different topological invariants, i.e., Chern numbers, robust edge modes localized around the interface between the photonic crystals emerge inside the bandgap [53,61,62]. Those edge states are topologically protected and are robust against defects and disorder, and allow transmission without any reflection, with no significant energetic loss [40,63,64]. Thus, in order to study the emergence of the edge states in our system, we built a supercell composed of 30 unit cells: 15 topological unit cells ($\phi = -\pi/4$) and 15 trivial unit cells ($\phi = \pi/4$). Next, we project the calculated band structure along the $\Gamma \to M$ direction as we can see in Figure 8. It is worthy to mention that, as usual in the literature, we consider simple harmonic electromagnetic waves, excited by a linearly polarized source, for calculating the propagating edge modes (see for example Refs. [6,44]).

From Figure 8, it is easy to identify two edge modes inside the gap and we notice that they emerge close to the $\Gamma$ point of the Brillouin zone. Those modes travel in opposite directions, and the traveling direction is reversed if we make $\mathbf{k} \to -\mathbf{k}$. In Figure 8b,c we show the profile of $E_z$, the phase of $E_z$, and the Poynting vector for $\omega a/(2\pi c) = 0.4349$ and $\omega a/(2\pi c) = 0.4383$, respectively. We used $\mathbf{k} = k_x \hat{x} = 0.015 k_0 \hat{x}$ with $k_0 = 4\pi/(\sqrt{3}a)$. Both modes are well localized at the interface and, by comparing $\omega a/(2\pi c) = 0.4349$ and $\omega a/(2\pi c) = 0.4383$, we see that the Poynting vectors have different directions, which is a confirmation of the pseudospin behavior of the edge states [18,65]. Moreover, focusing on the phase of the electric field $E_z$, we can identify that the mode with $\omega a/(2\pi c) = 0.4349$ presents a clockwise polarization, while the mode with $\omega a/(2\pi c) = 0.4383$ presents an anticlockwise polarization, corresponding to a pseudospin-down and pseudospin-up, respectively. Therefore, the pseudospin-up is associated with the interface state with group velocity and energy flux from the left to the right, while the pseudospin-down is associated with the interface state with group velocity and energy flux from the right to the left [65].

In the next section, we study the robustness and localization of the edge modes in our photonic system.

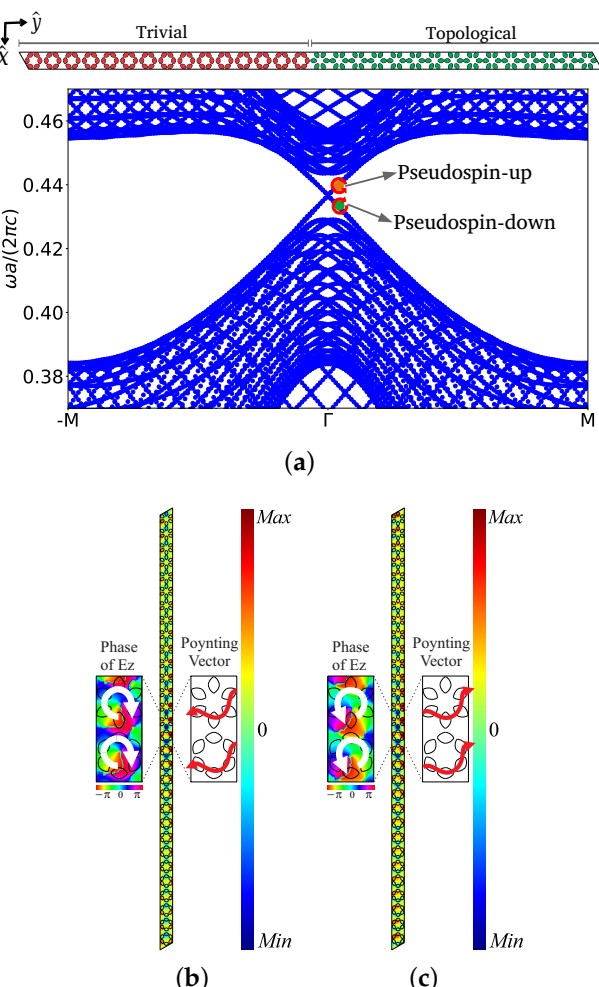

**Figure 8.** (**a**) Projected band structure along the $\Gamma \rightarrow M$ direction for TM modes (in blue) of a supercell composed of 15 topological unit cells ($\phi = -\pi/4$) and 15 expanded unit cells ($\phi = \pi/4$) making a horizontal interface. (**b**,**c**) $E_z$, the phase of $E_z$ illustrated by the white arrows (the **left panel**), and the Poynting vector illustrated by the red arrows (the **right panel**) for $\omega a/(2\pi c) = 0.4349$ (**b**) and for $\omega a/(2\pi c) = 0.4383$ (**c**) at $k_x = 0.015k_0$.

## 5. Robustness of the Edge States

The topological protection guarantees that the propagating modes, associated with the edge states, have good robustness against defects, disorder, and reflection at the interfaces [66–68]. So, in order to verify the edge states' robustness, we build a $(35, 20)a$ slab with a horizontal interface between topological ($\phi = -\pi/4$) and trivial ($\phi = \pi/4$) photonic crystals, respectively. We set a source of light on the left and a detector on the right, as shown in Figure 9a. Then, we calculate the normalized electric field defined as $E_N = \sqrt{|E_z|^2}$, shown in Figure 9b, for $\omega a/(2\pi c) = 0.4353$. Next, we introduce small defects at the interface: (i) a small cavity removing a rod (Figure 9c), (ii) a bigger rod of size $d_0 = 0.3d$ (Figure 9e), (iii) a Ag rod (the yellow one in Figure 9g), and (iv) a disorder at the interface changing one negative perturbed unit cell for one positive perturbed unit cell (Figure 9i). We also introduce extensive defects: (i) a Z interface (Figure 10a), and an (ii) Omega (interface Figure 10c). The Normalized Electric Field $E_N$, corresponding to the defects mentioned above, is illustrated in Figures 9b,d,f,h,j and 10b,d, respectively. Besides the normalized electric field, the figures also show the Poynting vector (red arrows) around

the defects, and around two points of the Z and Omega interfaces, as we can see in the zoom windows.

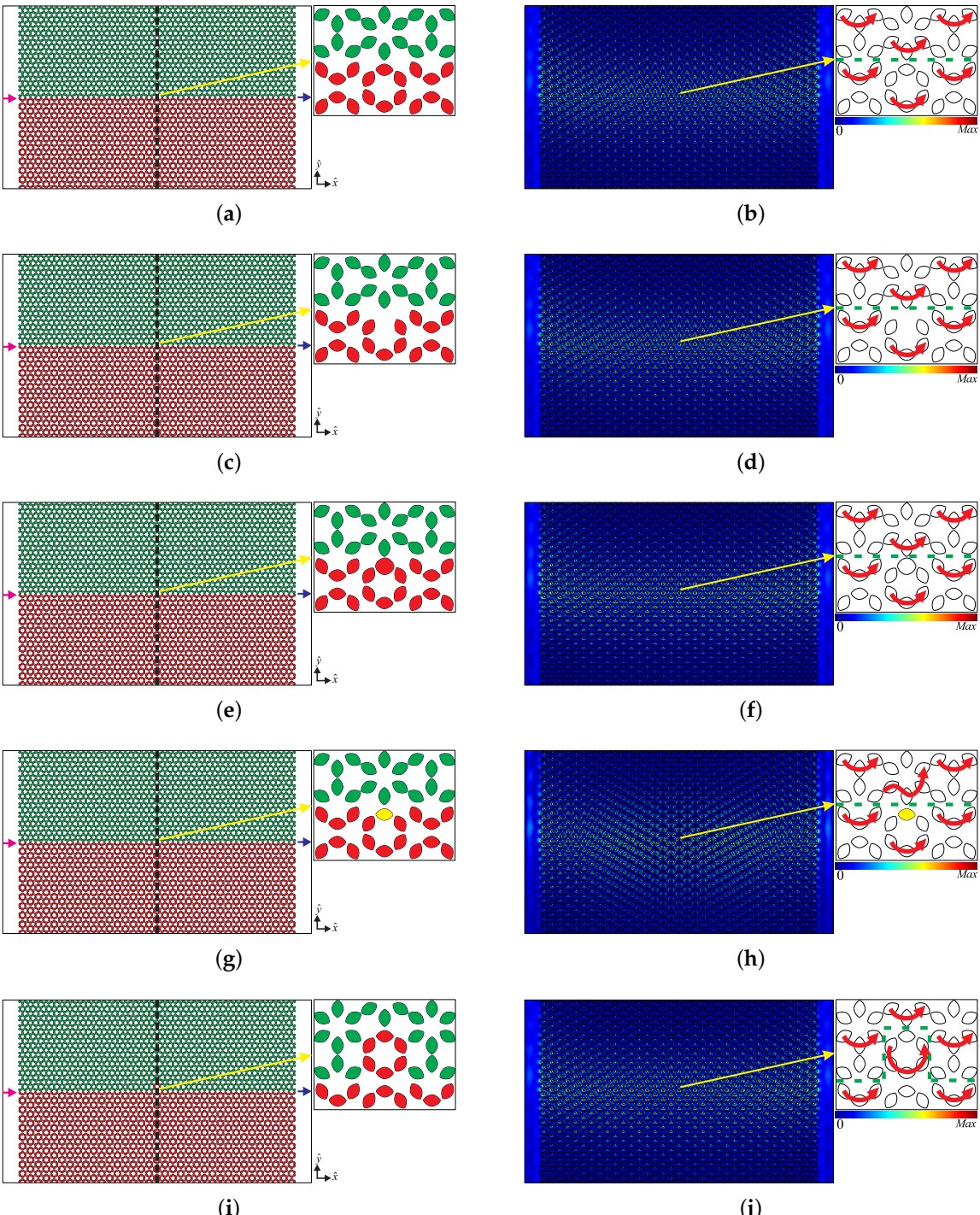

**Figure 9. Left panels:** schematic illustration of the interface between topological and trivial photonic crystals, $\phi = -\pi/4$ and $\phi = \pi/4$, respectively. The source of light is on the **left** (pink arrow) and the detector of light is on the **right** (blue arrow). We build an (**a**) interface with no defect, (**c**) interface with a small cavity, (**e**) interface with a bigger rod ($d_0 = 0.3d$), (**g**) interface with an Ag rod (the yellow one), and (**i**) interface with a disorder. **Right panels:** distribution of the normalized electric field $E_N$ and Poynting vector (red arrows in the zoom area around the defects) for $\omega a/(2\pi c) = 0.4353$ for the (**b**) interface with no defect, (**d**) interface with a small cavity, (**f**) interface with a bigger rod, (**h**) interface with an Ag, and (**j**) interface with a disorder.

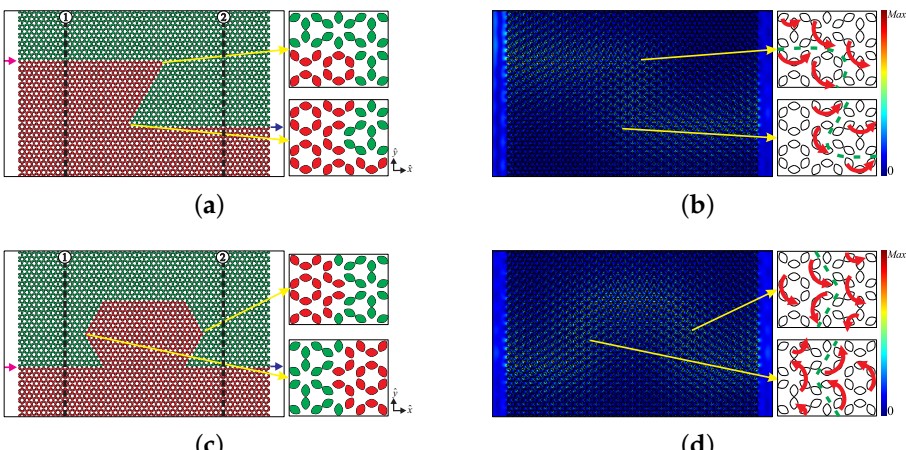

**Figure 10.** **Left panels:** schematic illustration of the interface between topological and trivial photonic crystals, $\phi = -\pi/4$ and $\phi = \phi/4$ respectively. The source of light is on the **left** (purple arrow) and the detector of light is on the **right** (blue arrow). We build a (**a**) Z interface and an (**c**) Omega interface. **Right panels**: distribution of the normalized electric field $E_N$ and Poynting vector (red arrows in the zoom area around the defects) to $\omega a/(2\pi c) = 0.4353$ to the (**b**) Z interface, (**d**) Omega interface.

For the pristine interface case, as expected, the electric field and the Poynting vector are localized around the interface, while the energy flux is from the left to the right (see Figure 9b). Furthermore, comparing Figure 9b,d, we realize that there is no significant change in the electric field around the cavity's position, and the Poynting vector is not captured by the cavity but just travels around it. Therefore, we can see in Figure 9d that the cavity created by removing a rod at the interface does not cause significant changes in the electrical field distribution and Poynting vector. Figure 9f corresponds to the bigger rod defect. Again, the presence of the defect does not cause changes in the edge mode, i.e., the electrical field distribution and Poynting vector are not affected. Despite the bigger rod, reflection does not occur and the energy flux is not affected by this defect at all. The electric field and the Poynting vector for the Ag rod defect are illustrated in Figure 9h. We could expect major changes in this case because of the energy losses involved. However, as in the previous cases, light does not experience significant changes. In fact, the energy losses are much smaller than the transmittance, as we will see later. On the other hand, we can observe local changes in Figure 9j. The disorder at the interface creates a different path for light. In this case, the energy flux locally changes around the defect, but the global behavior does not change, i.e., the energy flux keeps flowing close to the interface and with no significant reflections. As the mode is localized around the interface, both the Poynting vector's direction and electric field's distribution deform to follow the new interface shape at the position of the defect. Thus, we realize that the electric field survives and remains localized around the interface despite any defect introduced in the interface. In addition, the Poynting vector just walks around the defect or ignores it, and the flux of energy remains unchanged. Much more impressive are the results for the extensive defects: the Z and Omega interfaces. For the Z interface case, the interface has two corners in which light faces two changes of direction that could cause reflections, but the Poynting vector and electric field just follow the interface's contour and do not present any reflection in the corners (see Figure 10b). The Omega interface case is a more complex extensive defect. In fact, it has six corners which means that light faces six changes of direction! Despite the six corners, once again the Poynting vector and electric field follow the Omega interface's contour and they do not present any reflection in the corners (see Figure 10d). Therefore, we can conclude that the edge states are robust against defects, disorders, and reflection. This is guaranteed by the topological protection due to the bulk-edge correspondence. The results illustrated in Figures 9 and 10 are in agreement with previous works which investigated edge modes in topological valley photonic crystals [69,70] and topological

pseudospin photonic crystals [35,42,71–74]. It is worthwhile to remark that very recent experimental works have validated theoretical results, similar to ours, from the literature (see, for example, Refs. [75–78]).

In Figures 9 and 10 is provided a very good qualitative piece of information on the topological protection and robustness of the edge modes. However, it is important to quantify the robustness of the edge modes. The quantitative information is provided by the calculation of the transmittance of light along the system. Thus, the calculated transmittance through the slabs schematized in Figures 9 and 10 is plotted in Figure 11. We can observe in Figure 11 that transmittance changes very little when we introduce small defects such as a cavity, a bigger rod, an Ag rod, and a disorder. In fact, the effect of the defects is indeed small, so that the transmittance remains around 1. It should be remarked that for the Ag rod defect we would expect some losses because of the metallic character of the defect. However, only minor changes occur in the transmittance which remains around 1. Let us discuss now the extensive defects. For the Z interface case, we can observe a peak around 0.9, which is a 10% reduction in the transmittance in relation to the pristine interface. Despite this reduction in transmittance, the peak corresponding to the edge mode survives. A similar behavior is observed for the case of the Omega interface. Therefore, we can conclude that the transmittance peaks survive for the edge mode despite the small or extensive defects introduced in the interface. In short, our results show that the robustness of the interface mode is guaranteed against small and extensive defects in the interface, which means that light travels along the interface without changes in its energy flux, without reflection, and with minimal energy losses. Similar results were found in the literature with interface somehow modified: the peak can be reduced but the transmittance of light is at least ≥0.8, which means that most parts of light can travel through the considered system without reflection or absorption [11,79].

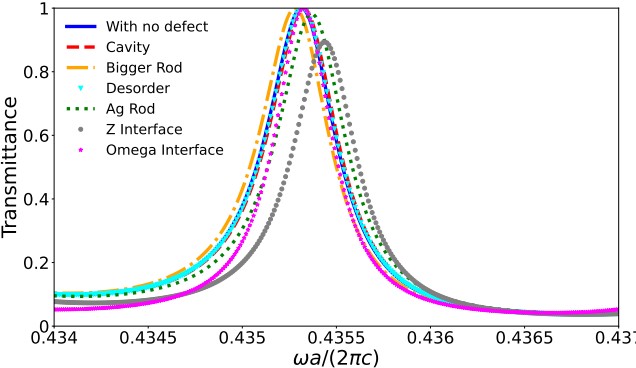

**Figure 11.** Transmittance for the cases without defect, with a small cavity, with a bigger rod, with a disorder in the interface, with an Ag rod, with a Z interface, and an Omega interface. We can observe that the transmittance is around 1.0 even for the configurations with defects.

Before concluding, after studying the robustness of the edge modes, let us take a look at the localization of the edge modes. In order to do so, we calculate $E_N / \max[E_N]$ point-by-point on a line perpendicular to the interface (the blue line in Figures 12 and 13). Therefore in Figure 12 is shown the intensity of $E_N / \max[E_N]$ point-by-point along the line of the interface with no defect and with small defects: the interface with a cavity, the interface with a bigger rod, the interface with a disorder, and the interface with an Ag rod. The same is illustrated in Figure 13 for the extensive defects: the Z interface and the Omega interface. We consider in all cases $\omega a/(2\pi c) = 0.4353$. From Figure 12a–e, we can infer that the electric field is localized around the interface, regardless of the small defect considered, i.e., the electric field is near zero in rods far from the interface for both cases: the topological and trivial photonic crystals. This corresponds to a topological insulator behavior of our system because it does not present fields in the bulk and the electric field is different from zero only around the interface [65,69,80]. For the special cases of extensive

defects, i.e., the Z and Omega interfaces, we set two perpendicular lines: (i) one at the interface before the first change of orientation, and (ii) another at the interface after the second change of orientation (see Figure 13a,b). As for the small defect cases, the interface modes are well localized. Notice that the normalized electric field rapidly goes to zero when the profile moves away from the interface for both the Z and Omega interfaces. In short, in all defect cases considered here, the localization of the modes allows the confinement of the light, and the interface works as a waveguide for the propagation of electromagnetic waves. Finally, all those features described here make this system a good candidate for topological wave guides and open up an opportunity for phototransport applications.

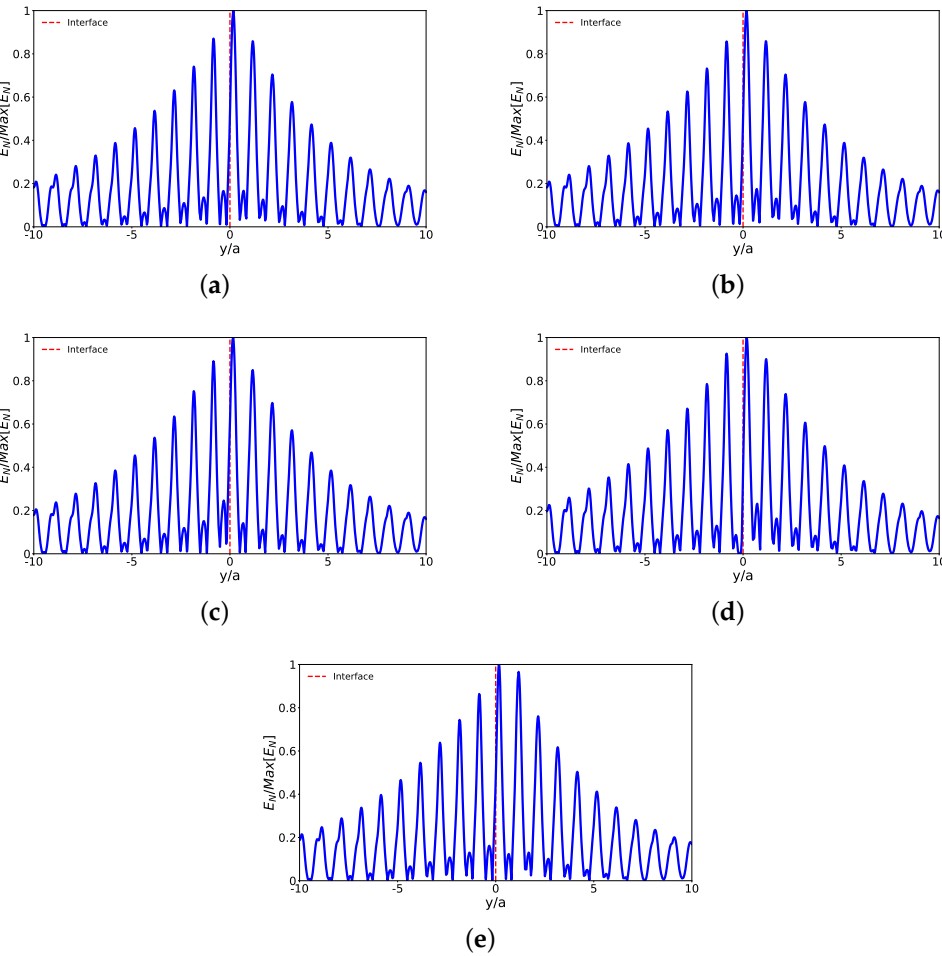

**Figure 12.** Intensity of $E_N / \max[E_N]$ vs $y$ ($\omega a / (2\pi c) = 0.4353$) for (**a**) interface with no defect, (**b**) interface with a small cavity, (**c**) interface with a bigger rod, (**d**) interface with an Ag rod, and (**e**) interface with a disorder.

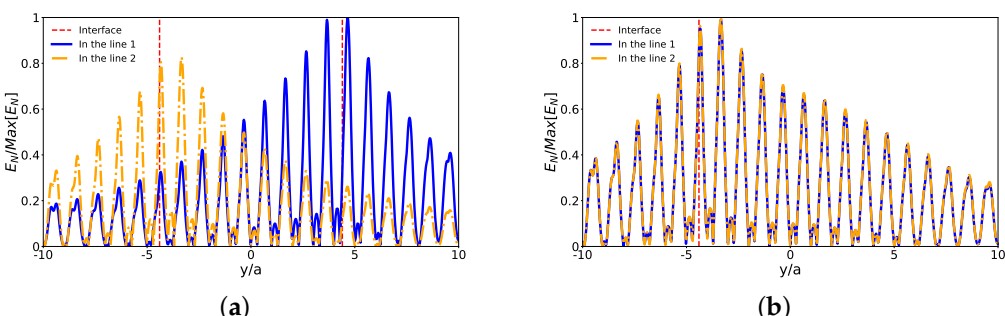

**Figure 13.** Intensity of $E_N / \max[E_N]$ vs $y$ ($\omega a / (2\pi c) = 0.4353$) for the (**a**) Z interface, and (**b**) Omega interface.

## 6. Conclusions

In this work, we have proposed a two-dimensional photonic crystal composed of dielectric cusped-oval-shaped rods (COS rods), in a triangular lattice with $C_6$ point symmetry group, with the unit cell composed of six COS rods. The band structure of the system was obtained through the software COMSOL Multiphysics, which is based on the finite element method (FEM). Our results show that, as we introduced an angular perturbation $\phi$ in the COS rods, a complete bandgap is opened in the band structure of the system. We have studied the topological phases of the system. It was found that for negative perturbations the photonic crystal is in a topological phase, while for positive perturbations the photonic crystal is in a trivial phase. The topological phase transition occurs for $\phi = 0$. We have also shown that the electromagnetic (EM) density energy distribution in the system can be used as a probe of the character of the band structure. This is because the EM density energy is proportional to the modulus square of the Wannier functions (WF) which are centered at the (maximal) localized Wyckoff positions (WP). Moreover, as a consequence of the perturbation, two edge states emerge inside the bandgap, which are localized around the interface and are topologically protected. Our results show that the edge modes are pseudospin modes traveling around the interface between the topological (negative perturbation) and trivial (positive perturbation) photonic crystals. They are topologically protected and robust against disorder and defects at the interface. For all defect cases considered in this work, the transmittance is mildly affected. Therefore, the system studied is an excellent candidate for technological applications, once the flux of light can be controlled without any significant energetic loss or reflection. These features make the photonic crystal here proposed a good candidate to guide and confine light, with potential for phototransport applications.

**Author Contributions:** Conceptualization, D.B.-S., C.H.O.C. and C.G.B.; methodology, D.B.-S., C.H.O.C. and C.G.B.; software, D.B.-S. and C.H.O.C.; formal analysis, D.B.-S., C.H.O.C. and C.G.B.; writing—original draft preparation, D.B.-S.; supervision, C.H.O.C. and C.G.B.; writing—review and editing, D.B.-S. and C.G.B. All authors have read and agreed to the published version of the manuscript.

**Funding:** This research was funded by Brazilian Research Agencies CNPq (Grant No. 309495/2021-0) and FUNCAP (Grant No. BP5-0197-00145.01.03/23). D.B.-S. acknowledges financial support from Brazilian Agency CAPES.

**Institutional Review Board Statement:** Not applicable.

**Informed Consent Statement:** Not applicable.

**Data Availability Statement:** The datasets generated during and/or analyzed during the current study are available from the corresponding author on reasonable request.

**Acknowledgments:** We thank G.M. Viswanathan for a critical reading of the manuscript.

**Conflicts of Interest:** The authors declare no conflict of interest.

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
