# Peer review of "Robust Topological Edge States in C6 Photonic Crystals"

_photonics, doi:10.3390/photonics10090961_

Round 1

Reviewer 1 Report

Photonics-2541503-comment

In this paper, the numerical results are presented regarding “ The unit cell of the system is composed of six rods organized on the sites of a C6 triangular lattice.” The authors induce a topological phase by introducing an angular perturbation  in the pristine system. The topology of the system is determined by using the so-called  perturbed mode. The manuscript is well presented and written, giving insights on the design and application of topology in photonic crystals. Here, I have some small comments on the results:

1. The content of the introduction should be enriched. “It is known that when the photonic crystal is in a nontrivial topological phase interesting phenomena can emerge”. Could the authors provide some examples of the interesting phenomena, the exotic properties, and the novel applications? The second paragraph of introduction should also be enriched. For instance, the authors may discuss the significant impact of the structural configuration of photonic crystals (involving the perturbation) on the formation of topology.

2. The key to the topological photonic crystal constructed in this article is to achieve band inversion. The authors could provide more details about Kramers degeneracy used in this work and add more references.

3. In Fig. 2a, we calculate the band structure for the TM modes (Ez, Hx, Hy 0) using the COMSOL Multiphysics software which is based on the finite element method (FEM)” . Could the authors provide the physical equations used in the simulation?

4. Band inversion for  and  orbitals are realized. These could be marked out in the band diagram in Fig. 4.

5. Figs. 8 b-c show two types of propagating edge states. Are these achieved by using a point-like chiral source or via some other means? Please provide some descriptions on this. Moreover, the positions of two modes illustrated in Figs. 8(b)-(c) and their corresponding spin directions should be marked in Fig. 8(a).

6. The electromagnetic transmission characteristics and the Poynting vector are presented in Fig. 9. Could the authors provide some experimental validations on this? If this is difficult to carry out, could you please provide some references related to the experimental measurements.

Author Response

Response to the reviewer’s comments is enclosed.

Reviewer 2 Report

In the present paper the authors propose the study of a two-dimensional topological photonic crystal with a unit cell composed of six rods organized on the sites of a C6 triangular lattice.

They show that the system presents a topological and a trivial phase, depending on the sign of an angular perturbation Ï• introduced in the pristine system.

They find and show typical results for topological photonic systems as the two edge modes at the interface between the trivial and topological photonic crystals.

They show that these modes are robust against defects, disorder and reflection. Moreover, they show that localization of the edge modes leads to the confinement of light and the interface behaves as a waveguide for the propagation of electromagnetic waves. They finally show that the two edge modes present energy flux propagating in opposite directions, the photonic analogue of the quantum spin Hall effect.

In my opinion, even if there is a lack of novelty in the results obtained, the work is worthy of publication since it is complete, clear, and pleasant to read.

I would only suggest the authors to better explain which is the role of the perturbation angle Ï•.

Is it related to some synthetic dimension?

Is it a modulation phase that gives an additional degree of freedom and permits one to relate the model with a higher dimensional “ancestor” as is the case of the Aubry-André-Harper model?

If it is the case which is the higher dimensional “ancestor” model of the one presented in the paper?

Finally, a comment on the fact that fig. 3 is not symmetric with respect to the perturbation angle could help.

Author Response

(The authors gave the same response as above.)

Reviewer 3 Report

This paper is well-written and well-organized. In this paper, you have studied the photonics system with C6 symmetry point group and shown that the edge states are robust in the interface between topological trivial and nontrivial photonic crystals with different defects. The protected edge states would improve the transportation of light if realized in practical use. 

Fig. 12 to add the data trace of the interface with no defect into the rest (b) (c) (d) (e) so the readers can have a direct idea about the influence of different perturbations.

It would make it even better with experimental data. 

Author Response

(The authors gave the same response as above.)

Round 2

Reviewer 1 Report

   The manuscript has been revised significantly according to the suggestions of the reviewer. It is well written and organized. I command for publication in present form.